# PETRA: PRETRAINED EVOLUTIONARY TRANSFORMER FOR SARS-CoV-2 MUTATION PREDICTION

## ABSTRACT

Since its emergence, SARS-CoV-2 has demonstrated a rapid and unpredictable evolutionary trajectory, characterized by the continual emergence of immune-evasive variants. This poses persistent challenges to public health and vaccine development.

While large-scale generative pre-trained transformers (GPTs) have revolutionized the modeling of sequential data, their direct applications to noisy viral genomic sequences are limited. In this paper, we introduce **PETRA**(**P**retrained **E**volutionary **TRA**nsformer), a novel transformer approach based on evolutionary trajectories derived from phylogenetic trees rather than raw RNA sequences. This method effectively mitigates sequencing noise and captures the hierarchical structure of viral evolution.

With a weighted training framework to address substantial geographical and temporal imbalances in global sequence data, **PETRA** excels in predicting future SARS-CoV-2 mutations, achieving a weighted recall@1 of 9.45% for nucleotide mutations and 17.10% for spike amino-acid mutations, compared to 0.49% and 6.64% respectively for the best baseline. **PETRA** also demonstrates its ability to aid in the real-time mutation prediction of major clades like 24F(XEC) and 25A(LP.8.1).

## 1 INTRODUCTION

Since its introduction to the human population, SARS-CoV-2 evolves at an extraordinary rate.(Amicone et al., 2022) Although the World Health Organization (WHO) stops designating new SARS-CoV-2 variants with new greek letters since 2022, novel Nextstrain major clades(Aksamentov et al., 2021), variants that possess significant additional mutations and achieve substantial regional or global prevalence at rapid speed, continue to emerge rapidly and persistently.

Such mutations often confer strong immune evasion against antibodies elicited by previous infections or vaccinations (Cao et al., 2022; Rubio-Casillas et al., 2022; Uraki et al., 2023), contributing to increasing rates of repeated infections and long-term sequelae. (Bowe et al., 2023) Consequently, the ongoing viral evolution and its health sequelae represent escalating societal challenges.(Al-Aly et al., 2024)

Simultaneously, the technology of Generative Pretrained Transformers (GPTs)(Radford et al., 2018) have undergone rapid development. Initially developed for natural language generation, GPTs have demonstrated remarkable effectiveness in handling various domains of sequential data including image, video, audio(Hurst et al., 2024) and genetic sequences(Avsec et al., 2025).

However, these models still struggle to model SARS-CoV-2 mutations. Genetic models are usually trained on DNA/RNA sequences. In genetic sequencing, the rate of sequence errors ranges from 0.4% to 13% depending on different sequencing methods(Dohm et al., 2020). This translates to 100-4,000 errors per sequence, far exceeding the number of mutations of SARS-CoV-2 variants.

In this research, we demonstrate the feasibility of training a generative pretrained transformer based on evolutionary trajectories extracted from phylogenetic trees instead of raw RNA sequences, reducing noise caused by artifacts during sequencing of individual sequences. We term the model **P**retrained **E**volutionary **TRA**nsformer(**PETRA**), and apply the model on SARS-CoV-2 mutation prediction.

Experimental results show that **PETRA** achieves significantly better performance in predicting future mutations compared to established baseline methods. Furthermore, **PETRA** demonstrates practical utility by aiding the real-time prediction of the evolutionary trajectories of major clades like 24F(XEC) and 25A(LP.8.1).

Our investigation into the global distribution of SARS-CoV-2 sequence data reveals substantial geographical and temporal imbalances, for which we develop a weighted training framework to address. While this approach can mitigate the effects of dataset imbalance, significantly enhancing the predictive performance of **PETRA**, we contend that the underlying data imbalance remains a significant challenge for comprehensive SARS-CoV-2 evolution tracking.

## 2 BACKGROUNDS AND RELATED WORKS

### 2.1 SARS-CoV-2 MUTATIONS AND PHYLOGENETIC TREE

Since SARS-CoV-2 starts to infect humans, people begin to collect its sequences. (Wang et al., 2020) SARS-CoV-2 is an RNA virus with a reference genome of 29,903 nucleotides. It contains multiple open reading frames (ORFs) that encode the proteins to be synthesized.

Mutations are changes to the nucleotide sequence, typically involving substitutions, insertions, and deletions. Some mutations within ORFs cause changes in the synthesized proteins, some even make large differences like frameshifts, early stops, or removal of whole ORFs, while others have no obvious functional impact. There exist some researches on the effects of some mutations (Plante et al., 2021; Motozono et al., 2021; Saito et al., 2022), but the exact functions of most mutations remain unknown. More recent studies show that the advantages of some of the most important mutations like S:Q493E on recent lineages rely on the existence of some other mutations.(Taylor & Starr, 2024) Therefore, different lineages may adopt different mutation patterns, trending towards mutation spectrum.(Bloom et al., 2023)

The Bloom estimator(Bloom & Neher, 2023) is an open-source deep-mutational-scanning(DMS) based project that scans through all mutations on every major variant and computes the likelihood to appear and potential fitness for mutation analysis and prediction. The project [1] is updated on a regular basis, but slowed recently. The latest version of the project is released in Nov 2024.

Separating different variants is vital given the mutation spectrum phenomenon. There are two variant classification systems, Pango(O'Toole et al., 2022) and Nextstrain(Aksamentov et al., 2021) for SARS-CoV-2. Nextstrain focuses on major clades that flourish on continental or global level, while Pango focuses on any novel lineages that could be related to possible epidemic events. Both systems are mostly maintained by a group of volunteer researchers who analyse sequences and discover and suggest new variants from time to time. The variant discovery process is now partially automated.(Zou, 2024) Details of the systems are introduced in the appendix.

Once a variant is designated, its relative growth can be estimated according to the sampling time and location of its sequences. New SARS-CoV-2 waves are usually driven by fast-growing new variants. Cov-spectrum(Chen et al., 2022) analyzes the growth of these new variants.

Most of the global SARS-CoV-2 sequences are primarily shared on the following platforms: GISAID(Shu & McCauley, 2017), GenBank(Benson et al., 2012), Cog-UK(Marjanovic et al., 2022), and CNCB(Members et al., 2024). Ultrafast Sample placement on Existing tRee (UShER) (Turakhia et al., 2021) aggregates sequence data on multiple databases to build a unified phylogenetic mutation tree. It maintains an existing mutation tree and attempts to add new sequences at the most likely positions, resulting in a continuously updated phylogenetic tree that represents the estimated mutation trajectories of all sequences.

Genetic sequences of SARS-CoV-2 are known to be imperfect since virus is introduced to the human population.(De Maio et al., 2020) To make things worse, the virus mutates and different variants start to co-infect people, expanding the source of artifacts. (Pipek et al., 2024) In genetic sequencing, the rate of sequence errors ranges from 0.4% to 13% depending on different sequencing methods(Dohm et al., 2020), which is 100-4,000 considering the reference length of 29,903 for SARS-CoV-2. On the other hand, SARS-CoV-2 variants have only 1-200 mutations in total . The extreme sequencing noise makes it difficult to train models directly on RNA sequences of SARS-CoV-2.

---

[1]https://github.com/jbloomlab/SARS2-mut-fitness

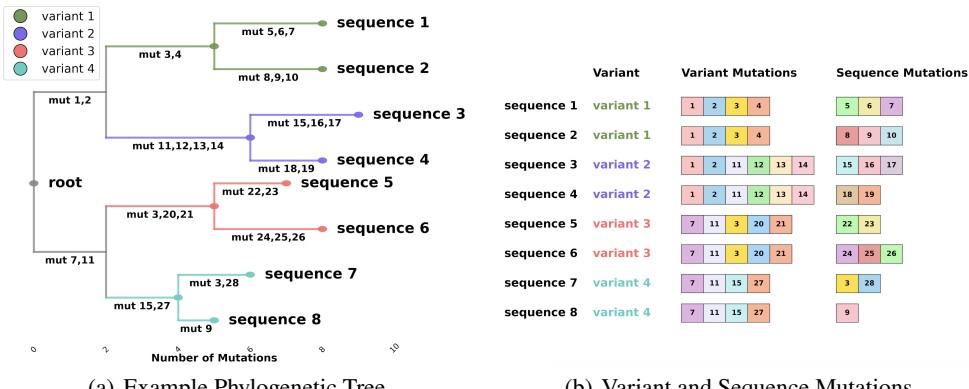

(a) Example Phylogenetic Tree  (b) Variant and Sequence Mutations

Figure 1: Left: An example phylogenetic tree. The virus starts from a root and gains mutations in sequential orders. Different branches may share some convergent mutations. Lineages with a sufficient number of mutations are designated as variants. Right: List of mutations for each sequence on the left phylogenetic tree. Each sequence has shared variant mutations and own private sequence mutations.

## 2.2 GPTs and their Applications on Genome Data

Transformers(Vaswani et al., 2017) is a type of attention-based neural network to handle sequential data. Generative pretraining is an unlabeled learning scheme that takes only raw sequential data and uses next token prediction using existing tokens as the training task. (Radford et al., 2018) Combining these methods results in GPTs, which revolutionized the modeling of sequential data, especially in natural language, and have become the major tool in modern artificial intelligence.

GPTs can also be applied on genome data. There exist powerful GPT-based models like Evo-2(Brixi et al., 2025) and AlphaGenome (Avsec et al., 2025) trained on natural DNA and RNA sequences. Despite being powerful, none of these models can be directly applied to SARS-CoV-2 mutation prediction tasks.

There also exist researches attempting to build up transformer-based models directly for SARS-CoV-2. (Zhou et al., 2023; Feng et al., 2024) Nevertheless, these attempts focus on specially framed datasets of sequences from certain countries and time periods, and are hard to generalize and update according to developments of the virus, making them practically useless.

To summarize, before **PETRA**, the best model that the variant tracking community relies on for SARS-CoV-2 mutation analysis and prediction is still the Bloom estimator mutation fitness model.

## 3 Methodology

### 3.1 Evolutionary Trajectories

According to physical principles, the evolution of viruses shall be step-wise. Every new variant shall either be a descendant of an existing variant or a recombinant of multiple existing variants. In physical terms, the evolutionary trajectory can be viewed as a tree where mutations are gained in sequential orders.

Although the actual evolutionary tree is unknown, there exist methods to estimate the evolutionary tree using existing sequence data. This task can be simplified to repeatedly estimating the placement of a new sequence on an existing tree. UShER maintains a phylogenetic tree and adds new sequences to placements with the minimal parsimony score on a daily basis.

This process does not automatically correct the errors in SARS-CoV-2 sequences, which sometimes distort the tree. The core contribution is from a research community [2] where voluntary researchers analyze sequences and report variants using the UShER system. They also report potential bugs and

---

[2] https://github.com/sars-cov-2-variants/lineage-proposals/

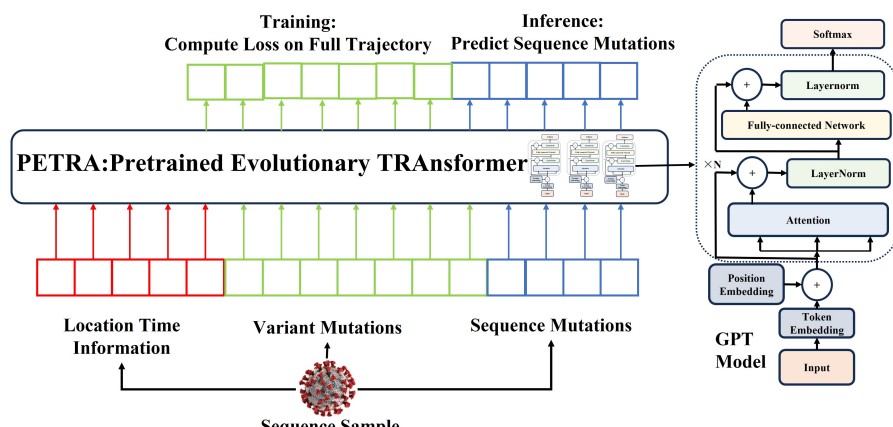

Figure 2: The training and inference of PETRA. Each Sequence is encoded to location time information, variant mutations and sequence mutations. During training, we compute loss on full evolution trajectory. During inference and evaluation, we predict the sequence mutations.

organize public discussions on how to fix them while using the system. Thanks to these precious human feedback, most of the errors are identified and manually fixed on the UShER tree.

Figure 1 offers an example phylogenetic tree. The virus starts from a root node, and evolves from time to time. Lineages related with potential epidemic events are designated variants. Each sequence belongs to a variant, shares variant mutations with a group of other sequences, and has its own sequence mutations on top of the variant mutations.

Although the error in UShER is largely reduced by the voluntary research group, it still has some systematic errors. For example, it does not model recombinants and regards them as having a sequence of mutations on top of one of its donors. We develop a process alleviating these errors through aggregating variant-level definitions of multiple platforms. Details of the process are described in Appendix.

## 3.2 The **PETRA** model

Figure 2 illustrates the structure of the **PETRA** model. The model is a decoder-only transformer with 116 million parameters. Each sequence is tokenized into three parts: the location time information, the variant mutations and the sequence mutations in sequential order. During training, the loss is computed on the full mutation trajectory including variant mutations and sequence mutations. During evaluation, the model only predicts sequence mutations on top of a variant.

The model uses a unified tokenizer of 150,210 tokens to tokenize location time information and mutation trajectories. Details of the model structure and tokenizer are presented in Appendix.

We train the model for 80,000 steps under a batch size of 256 using Adam optimizer(Kingma, 2014) with $\beta = (0.9, 0.95)$. The learning rate starts from 0.0001 and decays to 0.00001 during the training process. Given the training set contains approximately 17 million samples, each sample is seen slightly more than once in average. However, due to weighted sampling, some of the sequences may be sampled multiple times and some may not be sampled.

## 3.3 Imbalance of SARS-CoV-2 Sequences

Globally, resources and policies for sampling and sharing SARS-CoV-2 sequences vary significantly by region. Figure 3 illustrates the sequence distribution according to their economic development status.(Long & Ascent, 2020) A key finding is that developing and least developed countries are severely underrepresented in the global corpus of SARS-CoV-2 sequences.

Meanwhile, SARS-CoV-2 mutates within human hosts. The more people it infects, the higher the chance it gains mutations, irrespective of the host's economic status. The real-world representativeness of mutations and variants is expected to be proportional to the population they infect.

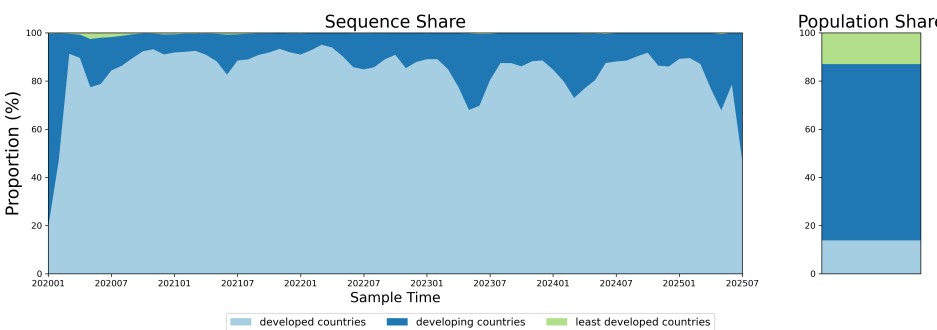

Figure 3: Distribution of SARS-CoV-2 sequences by country type. Developing and least developed countries are seriously underrepresented.

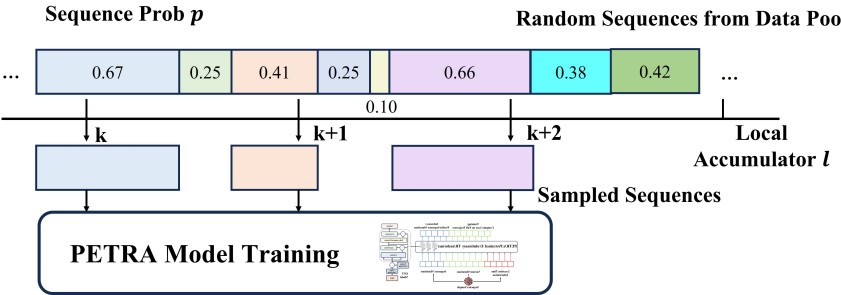

Figure 4: Distributed weighted sampling process of PETRA. Each worker maintains a local accumulator $l$. For each random sequence received from the data pool, it accumulates the sequence's probability $p$ to $l$. Only sequences that make $l$ meet or exceeds the next integer are selected.

To address this problem, we use a weighted system that weights the representativeness of each sequence according to their sample place and time. Although it is hard to estimate real-world infection levels due to shortness of data, we take the assumption that the actual scale of infection is proportional to the population and time.

Let $P$ be the population of a region, $n$ be the number of sequences submitted in that region in a certain time window, $d=\frac{n}{P}$ is the sample density. We develop a representative weight $r$ for each sequence based on sample density. Details of the representative weight $r$ are described in the Appendix.

In order to make use of the sequence weights, we develop a weighted sampling process during training. We set the sampling probability $p$ for a sequence in each epoch proportional to the logarithm of its representativeness $r$.

$$p = \frac{1}{m}(log\left(\frac{r}{r_0}\right) + 1) \qquad (1)$$

We ensure correct proportional sampling across distributed workers using a local accumulator technique. Each worker process maintains a local accumulator variable $l$. As a worker iteratively retrieves sequences from the shared data pool, it accumulates their probability values $p$ into $l$. A sequence is selected for the current training epoch if and only if adding its probability $p$ causes the local accumulator $l$ to meet or exceed the next integer. Figure 4 displays the training process.

Furthermore, because SARS-CoV-2 mutations are heavily influenced by host immune backgrounds which change rapidly over time due to immune imprinting and repeated infections(Yisimayi et al., 2024), we also apply a temporal reweighting for sampling probabilities of each training sequence.

$$p' = p(t_0 - t)^\lambda \qquad (2)$$

$t$ is the sample time(in month) of the sequence and $t_0$ is the release date of the training set.

|  | Sequences | Collection Date | Release Date |
|---|---|---|---|
| Total Sequences | 16,990,639 | All sequences | Before 2025-7-16 |
| Training Sequences | 16,772,588 | All sequences | Before 2025-2-12 |
| Evaluation Sequences | 57,117 | After 2025-2-13 | Before 2025-7-16 |
| Evaluation Sequences(Spike) | 15,137 | After 2025-2-13 | Before 2025-7-16 |

Table 1: Statistics of **PETRA** training and evaluation datasets. The model is trained on 16,772,588 sequences released before 2025-2-12. We evaluate the model on 57,117 sequences collected after 2025-2-13 and released before 2025-7-16.

## 4 EXPERIMENTS

### 4.1 TRAINING AND EVALUATION DATASET

We use different versions of UShER phylogenetic trees to train and evaluate PETRA. UShER updates the tree once a few months using all the data from different platforms released before the update. We use all of the 16,772,588 sequences on UShER tree updated on 2025-2-12 for training, and the sequences on UShER tree updated on 2025-7-16 for evaluation.

Each sequence has a collection date and a release date. The collection date is the date the sequence is sampled from a patient. The release date is the date it is released on any of the platforms. The gap between the release date and the collection date may be as short as a few days, or as long as multiple years.

The same sequence may be released to different platforms at different times. To avoid potential data leakage, we only use sequences that are collected after the release date of the training set in evaluation, as described in Table 1. We also freeze the existing variants and do not include any variants designated after 2025-2-12 for training.

We evaluate the model on the prediction accuracy of sequence mutations, the mutations a sequence further gains on top of a designated variant. We perform two prediction tasks: nucleotide mutation prediction(predicting the next nucleotide mutation), and spike mutation prediction(predicting the next spike amino-acid mutation). There are 57,662 sequences that were released before 2025-7-16 and collected after 2025-2-13. Among these sequences, 57,117 have at least one private nucleotide mutations on top of the variants they belong to, and 15,137 have at least one spike amino-acid mutations. Our evaluation sets for the two tasks are consisted of these sequences respectively.

Note that the variant definitions and estimated mutation trajectories of the same sequence may be different on the training and evaluation dataset due to the tree being optimized from time to time according to new sequence data. We use the variant definitions and mutation trajectories on the training set for training and the variant definitions and mutation trajectories in the evaluation dataset for evaluation.

### 4.2 BASELINE METHODS

Before this research, the variant tracking community usually uses the Bloom fitness estimator(Bloom & Neher, 2023) for mutation analysis and predictions. The estimator estimates the expected count and the fitness score of each potential sequence over major clades. It releases its estimations every several months. The last update is on 2024-11-6. We take the estimations from its latest update.

We take the mutations that score the highest to be its predictions. We find that using either expected count $c$ or fitness score $f$ does not yield as good predictive results as using a mix of the two scores, $s = ce^{\alpha f}$. We use $\alpha = 1$ and report the predictive result of all three scores.

### 4.3 EXPERIMENTAL RESULTS

Table 2 presents the prediction performance for nucleotide mutations. We report Recall@$k$ for $k = \{1, 10, 100\}$ on the task of predicting sequence mutations for all 57,117 sequences in the evaluation set. For sequences with multiple mutations, a sequence-level recall is first computed as the mean recall across all mutations within its trajectory. The overall performance is then reported in two ways: (1) as the macro-average (the direct mean of all sequence-level recalls), and (2) as the

| Method | Average Recall | | | Weighted Recall | | |
|---|---|---|---|---|---|---|
| | @1 | @10 | @100 | @1 | @10 | @100 |
| Random Guess | 0.00% | 0.01% | 0.08% | 0.00% | 0.01% | 0.08% |
| Bloom-Expected Count | 0.01% | 0.04% | 0.74% | 0.01% | 0.05% | 0.73% |
| Bloom-Fitness | 0.17% | 0.86% | 3.70% | 0.17% | 0.84% | 3.70% |
| Bloom-Mixed | 0.45% | 1.50% | 9.15% | 0.49% | 1.48% | 9.41% |
| **PETRA** | **11.34%** | **16.92%** | **22.64%** | **9.45%** | **14.20%** | **19.72%** |

Table 2: Nucleotide mutation prediction results for **PETRA**. We report average and weighted recall@1,10 and 100. In weighted measure, sequences are weighted by their representativeness.

| Method | Average Recall | | Weighted Recall | |
|---|---|---|---|---|
| | @1 | @10 | @1 | @10 |
| Random Guess | 0.01% | 0.13% | 0.01% | 0.13% |
| Bloom-Expected Count | 0.00% | 0.22% | 0.00% | 0.22% |
| Bloom-Fitness | 2.29% | 8.15% | 2.20% | 10.04% |
| Bloom-Mixed | 6.26% | 12.63% | 6.64% | 13.08% |
| **PETRA** | **17.84%** | **25.69%** | **17.10%** | **25.58%** |

Table 3: Spike amino-acid mutation prediction results of **PETRA**. We report average and weighted recall@1 and 10. In weighted measure, sequences are weighted by their representativeness.

weighted average, where each sequence's contribution is weighted by its representative score $r$ to mitigate the impact of sample imbalance.

There are 29,903 nucleotide sites in SARS-CoV-2, each has 5 possible status, A, T, C, G and deletion. A mutation changes a nucleotide from one state (A, T, C, G) to another or to a deletion. With 29,903 sites, this results in approximately 120,000 potential nucleotide mutations.

Although the absolute recall values seem not high, both the expected count and fitness scores of Bloom estimator perform much better than random guess. The fitness score yields better prediction results than expected count, and the mixed scoring performs better than using the fitness score only.

**PETRA** shows a substantial improvement over the Bloom estimator, improving weighted recall@1 from 0.49% to 9.45%. This order-of-magnitude gain suggests that SARS-CoV-2 mutations do follow predictable patterns in some sense, which can be captured by GPT models via a carefully designed training process.

Table 3 shows the prediction results of spike amino-acid mutations on 15,137 sequences that have sequence-level spike amino-acid mutations in recall@$k$, $k = \{1, 10\}$. Bloom estimator offers a spike amino-acid expected count and fitness score table and we use that table directly for prediction. For **PETRA**, we still let the model predict the nucleotide mutation and judge by whether the predicted nucleotide mutation can perform the targeted spike amino-acid mutation.

There are 1,273 amino acids on spike. Depending on the mutation trajectory, there are around 7,700 potential amino acid mutations via a single nucleotide mutation. The Bloom estimator performs better on spike-only prediction and achieves 6.64% recall@1 under the mixed scoring. **PETRA** lifts it to 17.10%.

These experimental results demonstrate that **PETRA** does offer a breakthrough over previous DMS-based methods on mutation prediction of SARS-CoV-2.

### 4.4 ABLATION STUDIES

#### 4.4.1 EFFECTS OF WEIGHTED SAMPLING

We examine the individual contribution of the temporal and representative weighted sampling to **PETRA**'s performance. Table 4 shows the performance of different **PETRA** variants using different

| Method | Weighted Training | | Nucleotide Recall | | | Spike Recall | |
| --- | --- | --- | --- | --- | --- | --- | --- |
| | Temporal | Representative | @1 | @10 | @100 | @1 | @10 |
| Bloom-Mixed | | | 0.49% | 1.48% | 9.41% | 6.64% | 13.08% |
| **PETRA**-NW | | | 3.04% | 5.85% | 13.33% | 8.87% | 26.13% |
| **PETRA**-TW | ✓ | | 8.94% | 13.85% | 19.59% | 17.00% | 25.72% |
| **PETRA**-RW | | ✓ | 4.25% | 7.64% | 15.15% | 11.32% | **29.58%** |
| **PETRA** | ✓ | ✓ | **9.45%** | **14.20%** | **19.72%** | **17.10%** | 25.58% |

Table 4: Weighted recall@$k$ for different variants of **PETRA** models depend on use of temporal and representative weighting during training. **PETRA**-NW does not use any weighted sampling methods, **PETRA**-TW uses the temporal weight only. **PETRA**-RW uses representative weight only.

weighted sampling methods during training. **PETRA**-NW treats all samples equally, **PETRA**-TW weights samples only according to their sample recency. **PETRA**-RW uses representative weights but does not refer to their sample recency.

**PETRA** performs slightly better than Bloom estimator even without using any weighting methods, demonstrating the effectiveness of evolutionary trajectory level pretraining on transformers. Both the temporal and representative weighted sampling independently improve the performance of the model. **PETRA** performs the best when both sampling methods are used.

### 4.4.2 Effects of immune background shift

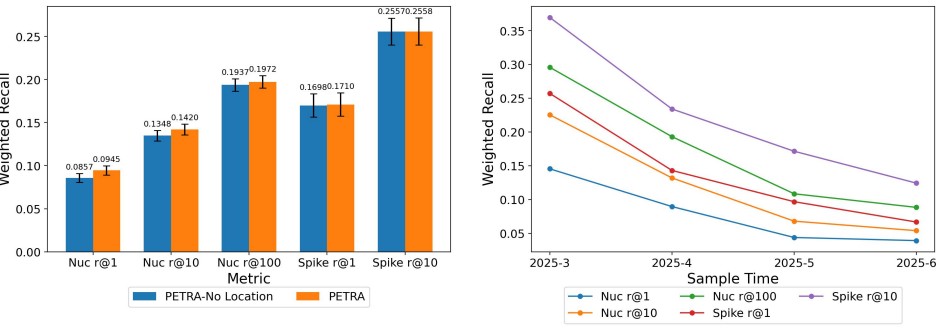

Figure 5: Performance of **PETRA** under different immune backgrounds. Left: Weighted recall@$k$ with and without location information. Right: Weighted recall@$k$ by different sample time.

Previous works have shown that immune background can affect viral evolution(Yang et al., 2024). Immune background is hard to measure due to shortage of data. However, there exist two dimensions of information that can indirectly affect the immune background: the geographical information and the temporal recency. Different regions have differences in circulating variants and follow different patterns of infection waves, resulting in different immune backgrounds. Population immune background is also changing over time due to new infections, vaccinations and immunity waning.

We display the result of two experiments in Figure 5. The performance of **PETRA** drops slightly when location information is not offered. Another observation is that the performance of the model decays substantially when evaluated on sequences sampled months after the training time.

Both observations demonstrate that the model does learn from the asymmetric distribution of different mutations under different immune backgrounds. The immune background varies more significantly over time than across geography. Therefore, we propose that **PETRA** be retrained once a few months using the most recent data to keep it at the best performance.

### 4.5 Real World Mutation Prediction

We also use **PETRA** to assist variant tracking researchers on real-world variant analysis and mutation predictions. During the past year, we have trained multiple **PETRA** models based on different updates

| Variant | Major Clade | Designation Time | Clade Elevation Time | **PETRA** Release Used |
|---------|-------------|------------------|----------------------|------------------------|
| XEC | 24F | 2024-8-7 | 2024-10-4 | 2024-6-6 |
| LP.8.1 | 25A | 2024-11-8 | 2025-1-25 | 2024-10-1 |

| Variant | **PETRA** Predictions | Designated Sub-variants | Undesignated Sub-lineages |
|---------|-----------------------|-------------------------|---------------------------|
| XEC | S:T572I | XEC.2.2.1, XEC.2.4, XEC.4, XEC.21, XEC.25.1.1 | 2 |
| | S:R346T | XEC.27 | 1 |
| | S:N185D | - | 2 |
| | S:A688V | - | 3 |
| LP.8.1 | S:A688V | QE.1 | 1 |
| | S:A475V | LP.8.1.9,NY.3.1.1,NY.6, NY.7.1.1,PD.1.1,PF.2.2.1,QH.1 | 9 |
| | S:T22N | LP.8.1.6,NY.7,NW.1.2,PP.1 | 0 |
| | S:Q677H | LP.8.1.3 | 1 |
| | S:H49Y | - | 0 |
| | S:G257S | - | 0 |

Table 5: Real world mutation prediction for Nextstrain major clades 24F(XEC) and 25A(LP.8.1). We use the **PETRA** model trained before the variant being designated to predict for their potential spike mutations and released our predictions in the variant tracking community. Most predictions match at least one sub-variants or undesignated tracked sub-lineages emerged lately.

of the UShER tree. When some of the fast-growing variants are designated, we use the latest **PETRA** model to generate and release the top spike mutation predictions to the variant tracking community, especially for XEC and LP.8.1, which are later elevated to Nextstrain major clades and become the global dominant variants in late 2024/early 2025 respectively.

Table 5 displays the predictions of **PETRA** and subsequent evolution for XEC and LP.8.1. We display both the designated Pango variants that are defined by the predicted mutations and the number of tracked sub-lineages with the predicted mutations but end up undesignated. Most of the predictions are followed by at least one designated sub-variant or undesignated tracked sub-lineages.

## 5 CONCLUSION

In this research, we propose **PETRA**, a pretrained evolutionary transformer that models the evolutionary trajectories of SARS-CoV-2 from phylogenetic trees. Pretrained on structured mutation pathways with temporal and representative weighted sampling, **PETRA** effectively captures the underlying patterns of viral evolution. Our experiments demonstrate that **PETRA** achieves a breakthrough in predictive accuracy, substantially outperforming the previous state-of-the-art DMS-based Bloom estimator. **PETRA** also manages to predict the mutation trajectories of newly-emerged real-world major clades in advance.

### 5.1 LIMITATIONS OF **PETRA**

Despite making decent progress, the complexity of SARS-CoV-2 mutations still poses great challenges to **PETRA**. Firstly, trained on the UShER phylogenetic tree, **PETRA** is only able to predict single nucleotide mutations but cannot handle recombination, which is playing a more and more core part in recent SARS-CoV-2 evolution. Secondly, **PETRA** is only able to predict mutations but cannot predict other features like severity, immune escape or growth potentials which public health departments may care about most. Finally, a core limitation is the imbalance of data, especially the shortage of sequences in developing and least developed countries. Many countries sample very few or even zero sequences for whole years, adding difficulty for **PETRA** to learn the diversity of SARS-CoV-2 mutations even with the weighted sampling approaches.

These limitations pose research challenges but also shed light for future developments of **PETRA**. A promising future direction is to enable the model to understand the effects of the predicted mutations better and reason through the imbalance of data.

**Ethic Statements**

PREVENTION OF DATA LEAKAGE    As detailed in section 4.1, we implement a rigorous temporal and data-source splitting strategy to prevent any data leakage between the training and evaluation sets. We apply a strict temporal split that only uses samples collected after the release date of the training set(2025-2-12), ensuring the model is evaluated on future evolution it has never seen. We also use distinct snapshots of UShER trees, preventing leakage from subsequent tree optimizations that might incorporate future knowledge. For evaluation, we used the variant definitions and mutation trajectories from the 2025-7-16 tree snapshot, simulating a real-world scenario where predictions are made based on the best available current knowledge.

These principles allow us to guarantee that our reported results reflect **PETRA**'s ability to forecast future mutations, rather than its capacity to memorize or retroactively fit to data that was inadvertently included from the future.

RISK OF ABUSE    As a generative model for mutation prediction, **PETRA** could theoretically be misused to generate hypothetical virus variants. However, the risk of such abuse is substantially mitigated by the model's inherent constraints. As discussed in section 5.1, **PETRA** is only able to predict the most likely natural mutations of the virus and lacks the capability to engineer novel variants with enhanced pathogenicity or transmissibility deliberately. Moreover, as shown in 4.5, the mutations forecast by **PETRA** are highly likely to emerge naturally. Therefore, the potential harm from malicious use is minimal compared to its benefit of providing early signals of future natural variants in advance.

USAGE OF GENERATIVE AI    We use generative AI to polish writing in this paper. Some of the figures are also generated via coding with generative AI. We use all the generated content for reference and write the words and code ourselves to ensure they represent the accurate meanings.

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

## A    APPENDIX

### A.1    MAJOR VARIANTS OF SARS-CoV-2

SARS-CoV-2 has undergone extensive mutations. Its actual evolutions form a very complex mutation graph, with 51 major Nextstrain clades and more than 5,000 Pango lineages.

The criteria for major Nextstrain clades are strict. The criteria have been updated multiple times and stabilized in April 2022. It requires a lineage to meet any of the following criteria to be designated a clade. [3]

- 1: A VOC or VOI is recognized by the WHO and given a Greek letter label.
- 2: A clade reaches more than 20% global frequency for 2 or more months.
- 3: A clade reaches more than 30% frequency for 2 or more months in any of the seven continents.
- 4: A clade shows consistent daily relative growth of more than 5% compared with other variants, and has reached 5% frequency in any of the seven continents.

These criteria ensure major clades must be significant variants that have either high prevalence or high growth. As they get their prevalence and growth from replacing previous variants, they serve as the key milestones of SARS-CoV-2 evolution. Nextstrain clades become especially important after WHO stopped recognizing variants with Greek letters.

Mutation Graph for Nextstrain Major Clades of SARS-CoV-2(As of Sep. 2025)

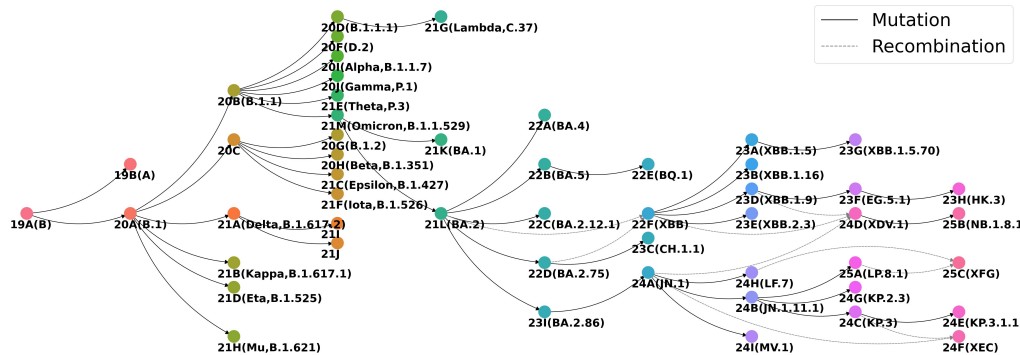

Figure 6: Simplified mutation graph of SARS-CoV-2 including nextstrain major clades. A variant must satisfy strict criteria to become a major clade. There are 51 major clades since SARS-CoV-2 was introduced to human population.

Figure 6 illustrates a simplified version of the mutation graph of SARS-CoV-2 containing the 51 Nextstrain major clades as of September 2025. Note that some of the major clades, such as 22F(XBB), 24D(XDV.1), 24F(XEC) and 25C(XFG) are formed via recombination of multiple variants.

### A.2    PANGO VARIANTS AND VARIANT-LEVEL DEFINITION REFINEMENT

Apart from Nextclade clades, Pango maintains a more relaxed variant system, where all lineages linked to a potential epidemic event can be designated a Pango variant.

The essential characteristics of a set of sequences to be designated a Pango variant is listed below. [4]

- 1: At the time of designation, the set of sequences is expected to share a single common ancestor and represent a monophyletic or paraphyletic clade in the SARS-CoV-2 phylogeny.
- 2: The clade should be distinguished by at least one unambiguous evolutionary event.

---

[3] https://nextstrain.org/blog/2022-04-29-SARS-CoV-2-clade-naming-2022
[4] https://web.archive.org/web/20240116214031/https://www.pango.network/the-pango-nomenclature-system/statement-of-nomenclature-rules/

- 3: The clade should contain a minimum of 5 sequences with high genome coverage.

- 4: The clade must include at least one internal node and therefore cannot be solely composed of a single polytomy. Thus, a lineage is expected to be consistent with a significant amount of onward transmission.

- 5: The clade should represent one or more events of epidemiological significance, including but not limited to the following events:

    5.1: The clade is a recombinant.

    5.2: The clade may represent inferred movement of the virus into a new country or region.

    5.3: The clade may distinguish successive epidemic waves in the same location.

    5.4: The clade may be observed to be growing rapidly and/or strongly increasing in frequency compared to other co-circulating lineages.

    5.5: The clade may be associated with observed or predicted changes in phenotypes including, but not limited to, transmissibility, immunogenicity, or pathogenicity.

    5.6: The clade may indicate a cross-species transmission event.

    5.7: The clade may carry a set of multiple mutations of particular biological interest or concern.

As of 2025-9-24, there exists 5,289 independent Pango variants of SARS-CoV-2. Most successful Pango variants will be upgraded to Nextstrain major clades.

There exists multiple platforms analysing SARS-CoV-2 variants from different perspectives. Nextstrain [5] from the emerging variant tracking perspective, UShER [6] from the phylogenetic tree building perspective, and Cov-spectrum [7] from the variant-level growth advantage tracking perspective.

The three platforms have their own version of variant mutation definitions and none of them is fully correct. They agree in most cases but disagree in a lot of corner cases. To achieve a better picture on defining mutations of variants, we leverage the three platforms and build a cross-validation data processing schedule.

- Step 1: For each variant, we first check the UShER tree and find the node that belongs to the variant and is closest to the root. We take its mutations as base.

- Step 2: We check Nextstrain definition and directly add insertions and deletions to the defining, as UShER does not contain such information.

- Step 3: For non-deleted codons that UShER and Nextclade disagree, we query Cov-Spectrum to see whether UShER and Nextclade is correct.

    Step 3.1: If a particular nucleotide state (including deletion) is present in more than 50% of the sequences for a given variant and appears at least 10 times more frequently than any other state, we treat it as part of the defining mutations for the variant.

    Step 3.2: If both sides does not satisfy the condition, we take the UShER definition.

This process aggregates variant definitions from multiple platforms, reducing potential errors of variant mutations to the minimal possible.

We follow the mutation trajectory on UShER tree and exclude mutations that are removed in step 3.1. Specifically, as UShER does not model recombinants, for designated recombinants, we list all of their mutations in position order and do not follow the order of the mutations on the UShER tree.

## A.3 DATA AVAILIABILITY

UShER maintains a phylogenetic tree including sequences from GISAID, Genbank, Cog-UK and CNCB. Genbank, Cog-UK and CNCB are fully public platforms while GISAID is a partial public

---

[5]https://nextstrain.org/staging/nextclade/sars-cov-2/
[6]https://genome-test.gi.ucsc.edu/cgi-bin/hgPhyloPlace
[7]https://cov-spectrum.org

|        | Mutations | Locations | Time | Special | Reserved | Total   |
|--------|-----------|-----------|------|---------|----------|---------|
| Tokens | 149,515   | 366       | 122  | 1       | 206      | 150,210 |

Table 6: Tokenizer of **PETRA**.

platform. According to user agreement of GISAID, anyone can register on GISAID, and GISAID sequences can only be distributed among registered users. [8]

We have registered accounts on GISAID, and we contact with the maintainers of UShER [9] to receive an updated UShER tree once a few months.

There is also a fully public version of UShER tree [10] that can be directly downloaded. This tree contains sequences only from the fully public platforms Genbank, Cog-UK and CNCB. However, the majority of the SARS-CoV-2 sequences are submitted to GISAID so this tree includes only a small fraction of total sequences. The fully public tree may also contain more errors than the GISAID tree, as it lacks the manual correction efforts provided by the variant tracking community, which primarily works with the GISAID data.

### A.4 DETAILS OF **PETRA**

#### A.4.1 TOKENIZER

We encode **PETRA** with a hand-made tokenizer of 150,210 tokens. There are 29,903 nucleotide sites in the reference sequence of SARS-CoV-2, each site has 5 potential states, A, T, C, G and deletion.

Following UShER, we encode potential mutations using only the mutated status of each codon and ignore their initial status. Mutations at the same codon with the same final status like A1T and G1T are treated the same in the encoder.

However, unlike UShER that shuffles the encoding for each token on every update, we unify the encoder and assign a fixed token for each of the $29,903 \times 5 = 149,515$ potential mutations. This change allows us to inference a **PETRA** model trained on previous UShER datasets on new UShER updates, or even manually uploaded new sequences.

For locations, we collect country and region level location information on UShER and uses 366 tokens to encode them. We also use 122 tokens to encode sample time, including 7 tokens for sample year, 84 tokens for sample months and 31 tokens for days of the month. We also use 1 special token to represent a lack of information for sequences without exact sample location or sample time. We reserve 206 tokens in our encoder for potential new locations and sample times in future dataset updates.

Table 6 displays the token distribution of **PETRA**.

#### A.4.2 STRUCTURE OF THE **PETRA** MODEL

|                   | Layers | Hidden Size | Attention heads | Max sequence Length | Parameters |
|-------------------|--------|-------------|-----------------|---------------------|------------|
| **PETRA**         | 12     | 512         | 8               | 2,048               | 116M       |
| **PETRA**-Large   | 24     | 1,024       | 16              | 2,048               | 458M       |

Table 7: Detailed parameters of **PETRA**. We also experiment a larger variant of **PETRA**, **PETRA**-large.

Table 7 displays the detailed parameters of **PETRA**. It is a generative pretrained transformer with 12 layers, 8 attention heads and a hidden size of 512. It has a max sequence length of 2,048 and uses RoPE(Su et al., 2024) position embeddings. It has 116M parameters. We train for 80,000 steps

---

[8]https://gisaid.org/terms-of-use/
[9]https://genome.ucsc.edu/contacts.html
[10]https://hgdownload.soe.ucsc.edu/goldenPath/wuhCor1/UShER_SARS-CoV-2/

| Method | Nucleotide Recall | | | Spike Recall | |
|---|---|---|---|---|---|
| | @1 | @10 | @100 | @1 | @10 |
| **PETRA** | 9.45% | 14.20% | 19.72% | 17.10% | 25.58% |
| **PETRA**-Large | **9.85%** | **15.35%** | **20.61%** | **17.48%** | **29.45%** |

Table 8: Weighted recall@$k$ for **PETRA** variants with different scales.

with a batch size of 256. Training can be completed with 8 nvidia 4090 GPUs in 12 hours using Megatron(Shoeybi et al., 2019) and flash attention(Dao et al., 2022) framework.

We also scale up **PETRA** to a larger setting of 458M parameters, with 24 layers, 16 attention heads and a hidden size of 1,024. Table 8 illustrates the experimental results when we scale up the parameters of **PETRA** from 116M to 458M. The model does perform slightly better as it scales up.

### A.5 WEIGHTED SAMPLING OF **PETRA**

We use a weighted sampling process during training. We compute two weights for each sample. The representative weight and the temporal weight.

#### A.5.1 REPRESENTATIVE WEIGHT

We compute sequence density $d$ by countries and regions for every month. $d = \frac{n}{P}$ which is the number of sequences $n$ divided by the population $P$ of a country or region in a certain month. We use country-level density for most countries, but use regional-level density of China, India and United States as they are the mostly populated countries on earth. We compute the sequence density of Chinese provinces, Indian pradesh and United States states independently. We use the estimated population in worldometers.info [11].

We use a density based representative weight to characterize the representativeness for each sequence. Given the population-based sequence density $d$ of a region at a certain time window described in section 3.3, we compute $r$ as the following.

$$r = \begin{cases} 1/\sqrt{d_0 d_1} & d \leq d_0 \\ 1/\sqrt{d d_1} & d_0 < d \leq d_1 \\ 1/d & d_1 < d \leq d_2 \\ 1/d_2 & d_2 < d \end{cases} \tag{3}$$

Our fundamental assumption is that the majority of SARS-CoV-2 mutations happen in human hosts, which is related more to the number of people who actually get infected rather than the number of sequences that are sampled or the development status of the country. As most parts of the world no longer have protective measures and embrace natural infections, the number of infections shall be proportional to the population, hence the representativeness of each sequence shall be proportional to the inverse of the sequence density $1/d$.

In some corner cases, some sequences are the only sequences from a large region at a certain time window, however they may only represent a smaller region inside the larger region, especially when $d$ being extremely small. For example, sometimes all the sequences from India only come from one hospital in Pune, Maharashtra, and only represent the variant distribution of the neighboring regions of Pune, which is a small fraction of people in India. Taking this into consideration, we set up a soft limitation that the representation starts deriving away from $1/d$ when $d$ is smaller than a fixed threshold $d_1$, and eventually bounded by $1/\sqrt{d_0 d_1}$. We also set up a representative lower bound of $1/d_2$.

In our experiments, we take $d_0 = 0.1, d_1 = 10, d_2 = 10,000$ sequences per million population per month. The representative score can be viewed as how much population a sequence represents in the month it is sampled. Each sequence has a base representative score of 100, and increases to 100,000 when the region samples only one sequence per 100,000 population in the month. The representative score further increases to 1 million when the sequence density is lower than 1 per 10 million. The score caps at 1 million.

---

[11]https://www.worldometers.info/world-population/population-by-country/

For $m$ and $r_0$ in equation 1, we take $m = 10$ and $r_0 = 100$.

### A.5.2 TEMPORAL WEIGHT

We also use temporal weight for weighted sampling during training. For $\lambda$ in equation 2, we take $\lambda = 0.1$.

| Method | $\lambda$ | Nucleotide Recall | | | Spike Recall | |
|---|---|---|---|---|---|---|
| | | @1 | @10 | @100 | @1 | @10 |
| **PETRA** | 0 | 4.25% | 7.64% | 15.15% | 11.32% | **29.58**% |
| | 0.05 | 7.66% | 12.17% | 18.05% | 14.68% | 27.33% |
| | 0.1 | **9.45**% | **14.20**% | **19.72**% | **17.10**% | 25.58% |

Table 9: Weighted recall@$k$ for **PETRA** variants with different temporal scaling factor $\lambda$.

Table 9 displays the performance of **PETRA** under different $\lambda$s. As can be seen, using $\lambda = 0.1$, the setting that prioritizes most recent samples, yields the best result in general.

### A.5.3 LIST OF DEVELOPED, DEVELOPING AND LEAST DEVELOPED COUNTRIES

| | |
|---|---|
| Developed Countries | Australia, Austria, Belgium, Bulgaria, Canada, Croatia, Cyprus, Czech Republic, Denmark, Estonia, Finland, France, Germany, Greece, Hungary, Iceland, Ireland, Israel, Italy, Japan, South Korea, Latvia, Lithuania, Luxembourg, Malta, The Netherlands, New Zealand, Norway, Portugal, Poland, Romania, Slovakia, Slovenia, Spain, Sweden, Switzerland, United Kingdom, United States |
| Least Developed Countries | Angola, Benin, Burkina Faso, Burundi, Central African Republic, Chad, Comoros, Democratic Republic of the Congo, Djibouti, Eritrea, Ethiopia, Gambia, Guinea, Guinea-Bissau, Lesotho, Liberia, Madagascar, Malawi, Mali, Mauritania, Mozambique, Niger, Rwanda, Senegal, Sierra Leone, Somalia, South Sudan, Sudan, Togo, Uganda, Tanzania, Zambia, Afghanistan, Bangladesh, Cambodia, Laos, Myanmar, Nepal, Timor-Leste, Yemen, Haiti, Kiribati, Solomon Islands, Tuvalu |

Table 10: List of developed countries and least developing countries from World Economic Outlook. Other countries are treated as developing countries.

Table 10 displays the exact countries we use for the developed, developing and least developed countries in figure 3. We refer to World Economic Outlook (Long & Ascent, 2020) for the categorization of developed, least developed and developing countries. We only list developed and least developed countries and treat all other countries as developing countries.

### A.6 DIRECT AVERAGE RECALL OF WEIGHTED SAMPLING SETTINGS

| Method | Weighted Training | | Nucleotide Recall | | | Spike Recall | |
|---|---|---|---|---|---|---|---|
| | Temporal | Representative | @1 | @10 | @100 | @1 | @10 |
| **PETRA**-NW | | | 3.44% | 6.44% | 14.38% | 8.23% | 25.34% |
| **PETRA**-TW | ✓ | | 9.07% | 14.51% | 20.99% | 15.63% | 24.43% |
| **PETRA**-RW | | ✓ | 4.69% | 8.48% | 16.06% | 10.62% | **27.97**% |
| **PETRA** | ✓ | ✓ | **11.34**% | **16.92**% | **22.64**% | **17.84**% | 25.69% |

Table 11: Direct average recall@$k$ for different variants of **PETRA** models depend on usage of temporal and representative weighting during training.

Table 11 displays the direct average recall@$k$ (without representative weighting in evaluation) for different variants of **PETRA** models depending on the usage of temporal and representative weighting during training. The results are similar to those of weighted recall@$k$ in Table 4.

