# OpenReview forum: "PETRA: Pretrained Evolutionary Transformer for SARS-CoV-2 Mutation Prediction"
_ICLR.cc/2026/Conference — Submitted to ICLR 2026_

### Official Review · Reviewer_vbe6 · 2025-10-22

**Soundness:** 2
**Presentation:** 3
**Contribution:** 2
**Rating:** 2
**Confidence:** 2

**Summary:**

The paper proposes PETRA, a transformer trained on evolutionary trajectories extracted from phylogenetic trees (UShER) rather than raw RNA sequence. They found PETRA can predict future SARS-CoV-2 mutations with a weighted recall@1 of 9.45% for nucleotide mutations and 17.10% for spike amino-acid mutations, which is a large improvement compared to 0.49% and 6.64% of the respective baselines.

**Strengths:**

The paper is overall well-structured and clearly written. It tackles an important problem of predicting SARS-CoV-2 evolution.

**Weaknesses:**

- The quality of the underlying phylogenetic trees data is questionable. The authors themselves acknowledge that the variant definitions from UShER, Nextstrain, and Cov-Spectrum “disagree in a lot of corner cases.”
- The sampling probability and temporal reweighting appears somewhat arbitrary, and it risk overfitting to recent and over-represented regions.
- Treating each country as homogeneous can distort the representativeness weighting and exaggerate biases toward well-sequenced urban centers.

**Questions:**

- The three-step variant-definition pipeline is reasonable, but appears heuristic. Have you evaluated the robustness of your method when trained on different tree construction methods?
- Any ablations to show sensitivity of sampling probability and temporal factor parameters?

---

> ### Author Response · Authors · 2025-11-17
> **Thanks for the review.**
>
> Thanks for the review.
>
> 1:For data quality issue, we acknowledge naturally-collected data shall have noise. These noise come from artefacts in sampling and sequencing. We also adopt sufficient methods to reduce the noise(for example, cross-validation of variant definitions across multiple platforms)
> Actually the main goal of data science and machine learning is to learn useful features from noisy natural data.  As experiments show, despite the noisy natural input, the PETra model does learn useful features that helps successfully predict the potential mutations of future SARS-Cov-2 mutations. Therefore, we strongly disagree with the judgement that noise from natural data is viewed as "major weakness" of the work.
>
> 2:  The sampling probability and temporal reweighting is designed to tackle the data imbalance problem in the input dataset. As Table 4 shows,  instead of "risk overfitting and over-represented regions", it reduces potential overfitting and regional over-representation issues and achieves better prediction results on when evaluated both on the reweighed and unreweighted metrics.
>
> 3: While we agree that treating each country as homogeneous may distort the representativeness weighting and exaggerate biases toward well-sequenced urban centers, there is no "urban centre" information available in GISAID or UShER dataset.  We use the fine-grained regional representation and uses regional information for most populated countries like China, India and the United States.
>
> 4: As for "different tree construction method", the UShER tree is constructed and manually fixed to represent the most likely natural mutation trajectory. There is no other "tree construction methods".
> Also notice that natural mutations are physical and unique, which means there exists an optimal tree that best represents the mutation trajectory of each sequences. Constructing the tree in other diverse ways will only introduce more potential errors.
>
> 5: Ablations to show sensitivity of sampling probability and temporal factor parameters are included in Table 9 and 11 in Appendix A.5.2 and A.6.
>
> We would like to emphasise that the sequences we use are not artificial data that are generated using some generation methods, but real-world viral sequences collected separate from global sequence labs and shared on GISAID and UShER platforms. While we do apply a lot of noise reduction data preprocessing methods like cross-validation on platforms, data reweighting, etc., the originality of the data is natural evolution, which we cannot control.

---

### Official Review · Reviewer_DEE8 · 2025-11-01

**Soundness:** 1
**Presentation:** 1
**Contribution:** 1
**Rating:** 0
**Confidence:** 5

**Summary:**

The authors trained a GPT model on top of existing sars-cov-2 sequences and try to predict the possible "next-strain" using the learned "evolutionary plausibility". While previous community effort on protein engineering have proven this strategy is effective, this paper is one of the several attempts in applying the same schema to dangerous infectious virus.

**Strengths:**

The paper is well written.

**Weaknesses:**

1. The model learns to extrapolate the pre-existing UShER tree, not viral evolution itself, making it useless for novel variants like Omicron where no such tree exists. It is pattern-matching on a graph, not learning biology.
2. The evaluation is critically flawed by the omission of direct comparisons to the actual state-of-the-art viral forecasting models discussed in recent scientific literature (e.g., scientific works mentioned in Nature News: https://www.nature.com/articles/d41586-024-04195-3).
3. The necessity of a massive GPT architecture is unproven, as the paper fails to benchmark against a much simpler, non-GPT autoregressive model applied to the same trajectory data.
4. Despite its predictive goal, the paper offers zero actionable scientific insights or generalizable rules of evolution, failing to justify the ethical risks of training a generative model on a dangerous pathogen.
5. The authors' ethical defense—that the model only predicts "natural" mutations—is doubly flawed: if so, why do we need your model? And the authors naively ignores the well-known risk of Transformer hallucination. Training generative models on viral sequences is fundamentally irresponsible.

**Questions:**

See comments above

**Details Of Ethics Concerns:**

This paper presents a fundamental and severe ethical problem that, in my view, makes it unsuitable for publication. The authors have trained a generative model, PETRA, with the explicit goal of predicting future, potentially more successful, mutations of SARS-CoV-2. This constitutes a form of in-silico gain-of-function research, providing a roadmap for engineering more dangerous viral variants. The authors' plan to release their models and code publicly is deeply irresponsible, as it would democratize access to a tool that could be misused for malicious purposes. The potential benefit of forecasting natural variants does not outweigh the significant and foreseeable risk of abuse. The entire line of research is ethically questionable and represents a dangerous precedent for the field of generative AI in biology.

---

> ### Author Response · Authors · 2025-11-13
> **Complete misunderstanding of the Paper**
>
> This review completely misunderstands the paper.
>
> 1: The USher tree is a tool to construct mutation trajectories for existing variants using aggregated data and manual fixes, while the model learns to predict natural mutation trajectories in future where no such tree currently exists. In fact, instead of the reviewer's claim "useless for novel variants like Omicron where no such tree exists", it is the most useful in the case of "novel variants where no such tree exists" and really helps real-world prediction and mutation trackings of novel major new variants like XEC(24F, late 2024) and LP.8.1(25A, early 2025) . (See discussions in https://github.com/sars-cov-2-variants/lineage-proposals/issues, issue 2088, 2199, 2597)
>
> 2: Instead of offering any practical baselines, the reviewer only offers a "Natural News" article to downplay the research. This is very unprofessional.
>
> 3: The reviewer mentions " a much simpler, non-GPT autoregressive model" without reference. Please offer practical examples to justify.
>
> 4: The reviewer claims "Despite its predictive goal, the paper offers zero actionable scientific insights or generalizable rules of evolution",  completely omitting the discussions in section 5 of the paper.
>
> 5: In the ethnic part, the reviewer describes the work as "in-silico gain-of-function research", this is a very weird term. The training data only contains existing natural mutation trajectories of the virus and the goal of the research is to predict conceding natural mutations that has not yet happened but may happen in the next few months to fuel for public health researches. It does not contain any functional data or "in-silico gain-of-function research".
>
> On the other hand, the reviewer also claims that natural mutations are not worth predicted in weakness 5, this strongly implies that he actually wants the model to predict non-natural mutations and considers the model only predicting natural mutations as a major weakness. This doubles down the flaw of his review.  It seems that it is the reviewer who is really keen on the unethical "gain of function research".

---

### Official Review · Reviewer_Zz8M · 2025-11-01

**Soundness:** 2
**Presentation:** 1
**Contribution:** 3
**Rating:** 2
**Confidence:** 3

**Summary:**

This paper proposes a transformer-based model, called PETRA, for learning the sequence of SARS-CoV-2 mutations which accumulate over time in different variants. A time- and geography- based weighting scheme is used to mitigate the effects of sampling biases and sampling time. The model can then be probed to predict the next mutations. On a new benchmark introduced in this paper, PETRA outperforms Bloom scores on predicting novel mutations of SARS-CoV-2.

**Strengths:**

- The time- and geography- based weighting is interesting and sounds more broadly applicable.
- The way in which the sequence is encoded is interesting -- one-hot encoding each site x mutation pair and concaternating them.
- Careful temporal train/test splits.

**Weaknesses:**

- The paper is poorly written in terms of grammar and phrasing. The abstract is split into three paragraphs. Grammatical errors are broadly present. Citation is non-standard, with citations after the period. An oddly harsh and dismissive phrase is used when referring to existing work:

"There also exist researches attempting to build up transformer-based models directly for SARS-CoV-2. (Shou et al., 2023; Feng et al., 2024) Nevertheless, these attempts focus on specially framed datasets of sequences from certain countries and time periods, and are hard to generalize and update according to developments of the virus, making them practically useless."

It is very likely that (1) this is the first time the authors are submitting to a major ML conference, (2) the authors are not fluent in English. These are not grounds for rejection, but it undermines the quality of the work.

-  The Bloom baseline is described as a "deep-mutational-scanning(DMS) based project" [side note: the space IS missing from the main text. There are several instances of these kinds of formatting oversights]. I took a look at the Bloom paper and it seems that Bloom is not a DMS-based method. The Bloom method used phylogenetic trees (much like PETRA) to map mutations and count their frequencies, leading to the fitness estimates for different mutations. The Bloom method is *validated* against DMS data, but is not a DMS-based method.
- PETRA only evaluates on its own mutation prediction task. How do the PETRA predictions correlate to the DMS datasets used in the Bloom paper? This would at least provide a clearer comparison against Bloom.
- It is not clear to me that Bloom scores are being used as intended by the Bloom paper. The PETRA paper proposes a composite Bloom score via s = ce^{\alpha f} which they show does better on their mutation prediction task than the Bloom fitness score or expected counts. This is quite odd. Why wouldn't the original Bloom paper propose such as score? Would this composite score s = ce^{\alpha f} also improve the Bloom correlations against DMS? Overall, it is not clear to me that the Bloom scores are being used as expected. Some discussion is necessary. The paper also arbitrarily sets \alpha=1 with no explanation.

**Questions:**

- Why did you split the abstract into three paragraphs?
- Why do you place citations after the period?
- Do you agree that the phrase "making them practically useless" is oddly harsh and dismissive of current published scientific work?
- Why do you call Bloom a "deep-mutational-scanning(DMS) based project"? From what I gathered from the Bloom paper, Bloom is a tree-based method (much like PETRA), which is *validated* against DMS data.
- Have you considered validating against the same DMS data as in the Bloom paper? While DMS data is not ground truth (several counterintuitive DMS scores are discussed in the Bloom paper), it would provide additional support for the performance of PETRA.
- How did you come up with the s = ce^{\alpha f} score for Bloom? How did you choose \alpha=1?
- Do you think the s = ce^{\alpha f} score for Bloom would improve correlation to DMS data?
- How exactly do you use Bloom scores to rank mutations? Provide further background on Bloom and how you use it in your benchmark.

---

> ### Author Response · Authors · 2025-11-12
> **Concern Regarding Irresponsible Reviewer Zz8M in Submission5014**
>
> This review is quite unprofessional to include baseless guesses like "this is the first time the authors are submitting to a major ML conference" and " the authors are not fluent in English." as part of the review.
>
> Despite the reviewer claims "The paper is poorly written in terms of grammar and phrasing", all examples he can provide is just some space and citation placement issues. It is irresponsible to judge a research article by artifactual rules like "the abstract shall only contain one paragraph" or "citations must be placed before the period" instead of its research values.
>
> Moreover, while claiming grammar flaws for the paper, the grammar issues in the review itself (for instance, using "Why did you" in Q1 while using "Why do you" in Q2) add ridiculousness to his judgement.
>
> We would like to publicly report this reviewer as "irresponsible reviewer" before rebuttal on the technical issues mentioned in the review.

---

> ### Author Response · Authors · 2025-11-19
> **Rebuttal of the Review**
>
> 1:For questions like abstract format or citation placement, we do not find rules included in ICLR submission guidelines that require the given format. Different people may have different versions of format requirements, but applying these personal issues to judgements is a signal of unprofessionalism.
>
> 2: As discussed in paper, these works focus on specific data formats that are hard to implement on new data. SARS-CoV-2 evolves rapidly and the mutation patterns change from time to time. Methods that can only be applied on fixed dataset that were created several years ago are not applicable to usage in real-world mutation tracking, which is the task PETra is aiming for.
>
> 3: Our model is built to predict real-world mutations, so we validate our model on the real-world mutation prediction task that directly predicts the next potential mutation given a sequence. Also DMS score data is only published to some past variants like XBB or JN.1, while PETra targets mutation predictions of new major clades like LP.8.1 and XEC and variants in 2025, which do not have DMS score data.
>
> 4: We evaluated multiple \alpha settings and it turns out using \alpha=1 gives the best predictive results for Bloom estimator.
>
> For the 2025-2-13~2025-7-16 evaluation dataset containing 57,117 sequences collected after 2025-2-13 and released before 2025-7-16 used in paper, use accuracy computed in the same way as in section 4.3 of the paper.
>
> \alpha=0(EC only): nuc pass@1,10,100: 0.01% 0.05% 0.73%  spike pass@1,10: 0.00% 0.22%
>
> \alpha=0.5: nuc pass@1,10,100: 0.49% 1.39% 8.89% spike pass@1,10: 6.64% 10.99%
>
> \alpha=1(used in paper): nuc pass@1,10,100: 0.49% 1.48% 9.41% spike pass@1,10: 6.64% 13.08%
>
> \alpha=1.5: nuc pass@1,10,100: 0.40% 1.27% 8.65% spike pass@1,10: 2.20% 15.65%
>
> \alpha=2: nuc pass@1,10,100: 0.24% 1.36% 8.27% spike pass@1,10: 2.20% 15.96%
>
> \alpha=\infty(fitness only): nuc pass@1,10,100: 0.17% 0.84% 3.70% spike pass@1,10: 2.20% 10.04%
>
> As can be seen, using \alpha=1 yields the best predictive performance on all nuc prediction metrics and Spike pass@1, only behind \alpha=1.5 or 2 on Spike pass@10.  So we consider \alpha=1 as the optimal choice for implementing Bloom estimator for mutation prediction.
>
> 5: For correlation to DMS data, actually the DMS data is not updated for most of the 2025 variants appeared in the evaluation set of the paper, so it is hard to decide whether it will improve or not.
>
> 6: For ranking metrics, we simply use s = ce^{\alpha f} on the most recent Bloom fitness score update for each potential mutation and rank mutations with the score s.

---

### Official Review · Reviewer_UrUX · 2025-11-01

**Soundness:** 2
**Presentation:** 3
**Contribution:** 2
**Rating:** 6
**Confidence:** 3

**Summary:**

This manuscript introduces PETRA, a pretrained evolutionary modeling framework designed to estimate virus mutations. PETRA leverages large-scale evolutionary sequence data to learn statistical patterns of antigen diversification, eliminating the need for experimental assays such as deep mutational scanning. The model outputs mutation-level fitness scores and functional annotations, and is intended to support high-throughput antigen evaluation and rational vaccine design. By learning both global evolutionary constraints and local biophysical signals, PETRA aims to generalize to previously unseen antigens and assist in predicting which mutations are likely to emerge or be tolerated under immune selection pressure.

**Strengths:**

A key strength of PETRA is that it addresses a major bottleneck in immunogen design: experimental characterization of mutational effects is slow, expensive, and inherently incomplete. The model’s zero-shot capability suggests it captures generalizable evolutionary principles rather than memorizing training examples. The authors emphasize that mutational fitness depends jointly on structural context and immune-driven selection, an important insight consistent with known antigenic drift dynamics. PETRA’s ability to annotate mutation functionality at scale is highly relevant for surveillance pipelines, early variant risk assessment, and computational vaccine candidate prioritization. The approach is also timely, given the growing interest in foundation-style models for biological sequence evolution.

**Weaknesses:**

One limitation is that the manuscript appears to focus primarily on sequence-based learning; explicit integration of three-dimensional structural context, epistatic coupling, or antibody–antigen interface geometry is not clearly articulated. Viral evolution is strongly epistatic, yet the evaluation setup seems to emphasize single-mutation effects, leaving open how well PETRA handles combinatorial variants observed in real variants of concern. The experimental validation section would benefit from broader benchmarking against state-of-the-art protein language models, sequence-to-fitness predictors, or phylogenetic fitness estimators. It is also unclear how robust PETRA is across diverse viral families with different evolutionary pressures. Finally, while zero-shot results are highlighted, the study does not deeply quantify failure cases, calibration, or false-positive risk.

**Questions:**

1. How does PETRA model non-additive interactions among multiple mutations, especially those common in real antigenic evolution (e.g., RBD co-mutational clusters)?
2. Does the model incorporate protein structure (e.g., contact maps, surface exposure), and if not, how might this limitation impact predictions at antibody epitopes?
3. How well does PETRA transfer to viruses with distinct evolutionary constraints, such as influenza HA or HIV Env?
4. Are fitness scores calibrated to biological magnitude (e.g., effect sizes comparable to deep mutational scanning measurements), or only relative rankings?
5. Can PETRA highlight specific residues or functional regions driving predicted fitness changes, and are these consistent with known neutralizing epitopes?
6. What controls are in place to avoid over-prediction of beneficial mutations, given that most random mutations are deleterious in nature?

---

> ### Author Response · Authors · 2025-11-19
> **Thanks for the Review.**
>
> We really appreciate the review that points out the main strength of PETra that we ourselves do not introduce well enough in paper, the early and low-cost inference of potential mutations before slow and expensive experimental characterization of mutational effects.
>
> For questions:
>
> 1: PETra can be inferenced on any given mutation checkpoint of the virus instead of designated variants or major clades only. This allows it to better model the combo effects of multiple mutations. Bloom estimator does give a good single-step prediction result under a major base variant and most of the performance gain of PETra over Bloom estimator shall come from better modelling of combo effects.  However, we have not specifically design experiments to evaluate exactly the scale for "multiple mutation prediction". The experiment shall be hard to design.
>
> 2: The model does not incorporate protein structures as we have not found a sufficient ways to incorporate them. Maybe it will get better but we have to first find ways to unify the protein structure data and mutation sequence data. It will be a great research topic.
>
> 3: We have not tested the performance of PETra on influenza HA or HIV Env. The framework shall be transferable but the access of training and evaluation data may be a limitation.
>
> 4: We only use fitness scores for relative rankings.
>
> 5: It can. As PETra offers a probability-based predictive result for each potential nucleotide mutation, we can sum up the nucleotide predictive probabilities related to specific residues or functional regions and use that probability as the prediction. We already predict the AA mutations and Spike mutations in this way, and the model does aid in the real-time spike mutation predictions for major clades like XEC and LP.8.1.
>
> 6: As our target is to predict natural mutations, we do not separate between beneficial and deleterious mutations. The model just predicts the next potential natural mutation.

---

### Meta-Review · Area_Chair_s8f1 · 2026-01-09

**Summary:**

This paper pretrains a transformer on evolutionary trajectories inferred from SARS-CoV-2 phylogenetic trees, with the goal being predict future mutations. While reviewers agree that the topic is timely, a majority of them raise concerns about baselines and writing, with scores (0, 2, 2, 6) and the only positive score 6 being pointed out by the previous AC to be potentially generated by LLMs (although it's not clear to me whether the reviewer simply used LLM for grammar polishing or for generating the entire review). Therefore, I do not recommend its acceptance in the current form.

**Reviewer Concerns:**

1. baselines and fairness of comparison: lack of stronger baselines such as specialized viral forecasting models.
2. lack of ablations on design choices such as data reweighting and sampling probabilities, which doesn't seem to be directly addressed in the rebuttal.

**Reviewer Scores:**

Given the strong opinions of the three reviewers who gave low scores (2 / 2 / 0), I think it is unlikely that they would change their scores by much if they had the chance to update their score.

---

### Decision · Program_Chairs · 2026-01-26

Reject